# Genome-Wide Characterization of the *INDETERMINATE DOMAIN* (*IDD*) Zinc Finger Gene Family in *Solanum lycopersicum* and the Functional Analysis of *SlIDD15* in Shoot Gravitropism

**DOI:** 10.3390/ijms251910422

**Published:** 2024-09-27

**Authors:** Huan Wu, Mingli Liu, Yuqi Fang, Jing Yang, Xiaoting Xie, Hailong Zhang, Dian Zhou, Yueqiong Zhou, Yexin He, Jianghua Chen, Quanzi Bai

**Affiliations:** 1School of Life Sciences, Division of Life Sciences and Medicine, University of Science and Technology of China, Hefei 230027, China; wuhuan113@mail.ustc.edu.cn (H.W.); zd134@mail.ustc.edu.cn (D.Z.); 2CAS Key Laboratory of Topical Plant Resources and Sustainable Use, CAS Center for Excellence in Molecular Plant Sciences, Xishuangbanna Tropical Botanical Garden, Chinese Academy of Sciences, 88 Xuefu Road, Kunming 650223, China; liumingli@mail.ynu.edu.cn (M.L.); fangyuqi@xtbg.ac.cn (Y.F.); yangjing3007@swfu.edu.cn (J.Y.); xiexiaoting@xtbg.ac.cn (X.X.); zhanghailong@xtbg.ac.cn (H.Z.); zhouyueqiong@xtbg.ac.cn (Y.Z.); h199911292022@163.com (Y.H.); 3University of Chinese Academy of Sciences, Beijing 100049, China

**Keywords:** *IDD* gene family, *SlIDD15*, gravitropism, *Solanum lycopersicum*

## Abstract

The plant-specific *IDD* transcription factors (TFs) are vital for regulating plant growth and developmental processes. However, the characteristics and biological roles of the *IDD* gene family in tomato (*Solanum lycopersicum*) are still largely unexplored. In this study, 17 *SlIDD* genes were identified in the tomato genome and classified into seven subgroups according to the evolutionary relationships of IDD proteins. Analysis of exon–intron structures and conserved motifs reflected the evolutionary conservation of *SlIDDs* in tomato. Collinearity analysis revealed that segmental duplication promoted the expansion of the *SlIDD* family. Ka/Ks analysis indicated that *SlIDD* gene orthologs experienced predominantly purifying selection throughout evolution. The analysis of *cis*-acting elements revealed that the promoters of *SlIDD* genes contain numerous elements associated with light, plant hormones, and abiotic stresses. The RNA-seq data and qRT-PCR experimental results showed that the *SlIDD* genes exhibited tissue-specific expression. Additionally, Group A members from *Arabidopsis thaliana* and rice are known to play a role in regulating plant shoot gravitropism. QRT-PCR analysis confirmed that the expression level of *SlIDD15* in Group A was high in the hypocotyls and stems. Subcellular localization demonstrated that the SlIDD15 protein was localized in the nucleus. Surprisingly, the loss-of-function of *SlIDD15* by CRISPR/Cas9 gene editing technology did not display obvious gravitropic response defects, implying the existence of functional redundant factors within *SlIDD15*. Taken together, this study offers foundational insights into the tomato *IDD* gene family and serves as a valuable guide for exploring their molecular mechanisms in greater detail.

## 1. Introduction

For living organisms on Earth, transcription factors are vital in gene regulatory networks, which are involved in regulating the growth, development, and environmental stress responses [1,2]. The IDD proteins are plant-specific Cys2His2 (C2H2) zinc finger transcription factor proteins, and the N-terminal of each contains two conserved C2H2 zinc finger structures and two C2HC zinc finger structures [3,4]. In addition, IDD proteins also have two variable domains in the C-terminal, namely the MSATALLQKAA domain and TR/LDFLG domain [5].

In the past decades, the role of IDD proteins in plants has been identified and extensively reported, with particular focus on the model species *Arabidopsis*, rice, and maize [6]. The first member identified in the *IDD* transcription factor family was *ZmID1*, which was reported to be involved in the regulation of the transition to flowering in maize via the autonomous floral inductive pathway [3,7,8,9,10]. Further studies have proved that *FLOWERING LOCUS T-like* (*FT-like*) *ZCN8* gene is placed downstream of *ZmID1* [11]. In rice, the gene *INDETERMINATE1* (*OsID1*)/*Early Heading Date 2* (*Ehd2*)/*RICE INDETERMINATE1* (*RID1*), which is homologous to *ZmID1*, contributes significantly to the transition from the vegetative stage to the reproductive stage [12,13,14,15]. *OsID1/EHD2/RID1* directly targets the *Heading date 3a* (*HD3A*) and *RICE FLOWERING LOCUS T1* (*RFT1)* to promote the reproductive stage of rice and can also act as a repressor to inhibit flowering through the *EHD1-HD3A/RFT1* pathway [15,16]. In *Arabidopsis* [3,13], *AtIDD8* regulates the photoperiod-dependent flowering pathway by modulating sugar transport and metabolism [17]. AtIDD3/MAGPIE (MGP) and AtIDD10/JACKDAW (JKD) physically interact with GRAS proteins SCARECROW (SCR) and SHORT-ROOT (SHR) and participate in the radial pattern regulation of roots [18,19,20]. The AtIDD10 and its homologous protein AtIDD9/BALDIBIS (BIB) functionally redundantly activate *SCR* expression to constrain SHR in the nucleus [19]. Other studies have shown that SHR and SCR can bind to the promoter region of *AtIDD8* and *AtIDD10*, and these transcription factors are required to participate in the refinement regulation of endodermal *SHR* specification [19,21,22]. *AtIDD6*/*BLUEJAY* (*BLJ*) and *AtIDD4*/*IMPERIAL EAGLE* (*IME*) are involved in organizing the ground tissue after embryogenesis [23]. In addition, AtIDD15/SHOOT GRAVITROPISM 5 (SGR5) and its homologous protein Loose Plant Architecture1 (LPA1) in rice and maize have been reported to influence aerial organ morphogenesis and gravitropic responses [24]. By regulating both light and hormonal signals, *AtIDD1*/*ENHYDROUS* aids in breaking the dormancy of seeds, enabling them to germinate rapidly under favorable environmental conditions [25]. *ZmIDDveg9*/*NAKED ENDOSPERM1* (*NKD1*) and *ZmIDD9*/NAKED ENDOSPERM2 (*NKD2*) have a major bearing on maize endosperm development and seed maturation, ensuring normal seed development and nutrient accumulation through the regulation of gene networks and cell fate determination [26,27]. Moreover, the DNA-binding domains of five IDD proteins (AtIDD3, AtIDD4, AtIDD5, AtIDD9, and AtIDD10) act as transcriptional scaffolding antagonists of DELLA proteins to regulate the expression of their downstream targets, thereby controlling the gibberellin (GA) signaling pathway [28]. Overall, these previous studies have shown that *IDD* genes are involved in various regulatory networks of plant growth and development.

To adapt to the habitat environment efficiently, plants sense the surrounding environmental cues, such as gravity, light, and temperature, to optimize their growth and development [29]. Gravity is recognized as a directional cue and a fundamental force under which all living organisms evolve. Once gravity stimulus is sensed by a plant, its shoot gradually grows upwards against gravity known as negative gravitropism [30]. In general, gravitropism affects plant architecture by regulating the growth direction of the lateral organs, thereby influencing the plant reproductive traits. Thus, gravitropism-influenced plant architecture is a key aspect for consideration for the genetic improvement of crops [31]. The stem negative gravitropism response process can be divided into three stages: in the first stage, the stem endodermal cells sense the gravity signal; in the second stage, signal transduction induces the auxin flowing towards the morphologically lower side; in the third stage, the polar distribution of auxin causes the elongation zone of the stem to grow towards the morphologically upper side [30,32]. Through the influence of the gravity sensing and auxin signal pathway, the *SGR5* is involved in the gravitropic response of *Arabidopsis* inflorescence stems [33]. In rice, *LPA1*, an ortholog of *AtDD15*, also influences shoot gravitropism and architecture [34]. *ZmLPA1* has also been reported to regulate leaf angle in maize via the auxin pathway [35].

Tomato is widely cultivated, highly productive, and one of the most popular fruits and vegetables globally. Due to its short growth cycle, high efficiency of genetic transformation [36], available genome information [37], and a variety of research traits, tomato has gradually become a model species and is extensively utilized in various studies [38]. At present, genome-wide identification of the *IDD* transcription factor family has been conducted in many species, including *Arabidopsis* [3], rice [5], maize [39], *Brassica napus* [40], *Malus* [41], and *Phyllostachys edulis* [42]. In addition, the functions of multiple *IDD* members in plant development have been well-characterized. However, information on the tomato *IDD* gene family and their functional analysis is rarely reported. In addition, although gravitropism plays a crucial role in shaping plant architecture and influencing reproductive yield, the underlying molecular mechanisms of gravitropism in *Solanaceae* plants have rarely been reported.

In this study, we identified the *SlIDD* gene family through bioinformatics analysis and performed phylogenetic tree analysis, chromosome localization, gene structure analysis, conserved motif identification, promoter *cis*-element prediction, and gene expression analyses. Subsequently, we conducted expression analysis on the tomato genes within Group A, which have been reported to participate in gravitropism regulation, and further performed functional analysis on *SlIDD15* utilizing gene editing technology. Our findings offer comprehensive insights into the *IDD* gene family in tomato, which will aid in exploring the functions and regulatory mechanisms of *SlIDD* family members.

## 2. Results

### 2.1. Identification and Physicochemical Analysis of IDD Members in Tomato

IDD proteins were identified in the tomato genome through BLAST and HMM searches. After manually checking and removing the candidate genes of incomplete ID domains based on sequence alignment, we identified 17 SlIDD proteins. These proteins were renamed SlIDD1 to SlIDD17 according to the identification by the ZmID1 protein sequence from highest to lowest (Table 1).

The physicochemical analysis revealed that the 17 SlIDD proteins had a range of 339 to 656 amino acids and had molecular weights between 38.6 kDa (SlIDD13) and 68.0 kDa (SlIDD17). The predicted isoelectric points (pI) of the proteins fell within a range of 8.24 (SlIDD6) to 9.38 (SlIDD4), conclusively categorizing them as alkaline in their physicochemical properties. The instability index of the 17 SlIDD proteins ranged from 39.99 to 79.82, and only SlIDD3 was less than 40, which belongs to stable proteins. The aliphatic index of the proteins was 53.55 (SlIDD1) to 69.38 (SlIDD13). The grand average of hydropathicity of all proteins was less than zero, indicating that SlIDDs were hydrophilic proteins. Furthermore, the predicted SlIDD proteins were all located in the nucleus (Table 1).

### 2.2. Chromosomal Location and Multiple Sequence Alignment of the SlIDD Gene Family

Multiple sequence alignment analysis showed that all SlIDD proteins had a conserved region of about 160 amino acids at the N-terminal, known as the ID domain (two C2H2 and two C2HC) (Figure 1a). In addition, we also analyzed the C-terminal domains of SlIDD proteins: the TR/L/QDFLG domain and MSATALLQKAA domain (Appendix A). We found that the C-terminal domains did not exist in SlIDD13 and SlIDD15 proteins. Then, according to the annotation information, we constructed the mapping of the 17 *SlIDD* genes in chromosomes (Figure 1b). The results showed that there was no distribution of *SlIDD* genes on Chr 5 and Chr 12. The *SlIDD* genes were mainly distributed on Chr 1 (two), Chr 2 (two), Chr 4 (two), Chr 6 (three), and Chr 9 (three), and the remaining five chromosomes each contained only one *SlIDD* gene. A majority of the *SlIDD* genes were distributed at the terminal of the chromosomes.

### 2.3. Conserved Motifs and Gene Structure of the SlIDD Gene Family

The 17 SlIDD protein sequences were analyzed for their evolutionary relationships, gene structure features, and conserved motifs (Figure 2). These can be divided into three subgroups by phylogenetic analysis. SlIDD13 and SlIDD15 were allocated to Subgroup 1, SlIDD6, SlIDD16, and SlIDD17 were allocated to Subgroup 2, and other SlIDDs were allocated to Subgroup 3. We analyzed the full-length protein sequences of the 17 SlIDD proteins with the MEME tool and identified ten conserved motifs (Figure 2 and Appendix A). Motif 1, motif 2, and motif 3 were present in all SlIDD proteins, constituting the N-terminal conserved ID-domain. Motif 4 and motif 6 were significantly absent from SlIDD13 and SlIDD15 in Subgroup 1 compared with other SlIDDs. This is consistent with the C-terminal protein multiple sequence alignment map of SlIDDs (Appendix A). Motif 8 was only present in SlIDD16 and SlIDD17 proteins in Subgroup 2. Motif 5 is a conserved nuclear localization signal sequence (KK/RK/RR), but motif 5 was absent from the SlIDD8 protein sequence. The results indicated similarity in function among SlIDDs within subgroups due to consistent motif composition and arrangement, whereas distinct motifs in different subgroups contributed to functional diversification within the family.

Concurrently, we constructed the exon–intron structure map. The *SlIDD* gene family members contained three to four CDS sequences, of which seven members contained three CDS sequences and ten members had four CDS sequences (Table 1). The *SlIDD8* gene had no untranslated region (UTR); however, the other 16 genes contained at least one UTR. In the process of evolution, the introns of genes may contain parts of genes that lose function and accumulate more mutations. For example, *IDD9*, *IDD12*, and *IDD16* all contained four introns, which may play a specific role in the evolution of tomato.

### 2.4. Phylogenetic Analysis of SlIDD Family

To gain a deeper understanding of the SlIDD protein from the perspective of species evolution, we constructed a phylogenetic tree that included the 71 IDD proteins and an outgroup protein, AtARF1 (Figure 3; Appendix A). It was found that the 71 IDD proteins were classified into a total of seven subgroups, namely Group A to Group G (Figure 3). Each group had at least one SlIDD protein. Apparently, the majority of SlIDDs were categorized into Group E, with seven SlIDD members, and there were no IDD proteins from monocotyledonous species present in this group. In addition, SlIDD3 and AtIDD12 were grouped separately into a clade in group B. Groups A, D, F, and G contained IDD proteins from four species, and it was observed that the SlIDDs in these groups clustered into a small subclade with *A. thaliana*. Phylogenetic relationships illustrated the variability of *IDD* genes between monocotyledonous and dicotyledonous plants. We found it interesting that in Group C, the ZmIDD7 and OsIDD7 formed a distinct clade with SlIDD2 and SlIDD5, and none of the AtIDD proteins were categorized. This may suggest that SlIDD2 and SlIDD5 might be more closely associated with the evolutionary development of monocotyledonous plants.

### 2.5. Gene Duplication and Evolutionary Analysis of SlIDD Gene Family

By exploring gene duplication within gene families through comparative genomics analysis methods, the diversity and expansion of these gene families can be analyzed in depth. We used MCScan X software to analyze genome-wide duplication events of *SlIDD* gene family members and mapped them in Circos (Figure 4a; Appendix A). The analysis identified ten duplicate gene pairs in *SlIDD* genes, all resulting from gene segment duplications. This result suggested that *SlIDD* genes may have undergone family expansion during evolution, primarily driven by segment duplication events. Ten gene pairs showed Ka/Ks ratios of less than one, indicating purifying selection in evolution (Table 2). Finally, we estimated the time of duplication events based on the Ks values, which showed that they occurred between 25.89 and 93.42 million years ago (MYA) (Table 2).

To delve deeper into *SlIDDs* evolution, we identified collinear genes in tomato and four species (*Arabidopsis*, rice, maize, potato) and plotted comparative covariance maps (Figure 4b; Appendix A). The results showed orthologous gene pairs between tomato and *Arabidopsis* (22), potato (36), rice (5), and maize (4). These results infer that *SlIDD* genes should be closely related to dicotyledonous plants during long-term evolution, and IDD proteins were highly conserved in two *Solanaceae* species. Additionally, *SlIDD1*, *SlIDD2*, and *SlIDD13* were found to have collinear gene pairs across all four species, indicating their irreplaceable role in the evolution of *SlIDDs*.

### 2.6. Analysis of Cis-Regulatory Elements in SlIDDs Gene Promoters

The promoter region of a gene is capable of regulating gene expression in response to both internal and external signals, and its *cis*-acting elements can effectively predict the function of the gene. We used the PlantCARE database to analyze the *cis*-regulatory elements of the 2000 bp promoter region upstream of its coding sequence. The *cis*-acting elements retrieved from the promoter region of the *SlIDD* genes were categorized into four groups, with a total of 52 types (Figure 5; Appendix A). There were 11 *SlIDD* genes that contained four types of *cis*-acting elements and six *SlIDD* genes that did not contain growth and development regulatory elements. The largest numbers identified were light-responsive elements, with a total of 23 types. All *SlIDD* genes contained 5–17 light-responsive elements, with box 4 elements present in all *SlIDDs* (Figure 5; Appendix A). Meanwhile, members of the *SlIDD* gene family also contained various hormone-responsive elements, including elements associated with auxin, gibberellin, abscisic acid, salicylic acid, and methyl jasmonate responses. There were ten *SlIDD* genes containing TGACG-motifs and CGTCA-motifs involved in MeJA-responsiveness, respectively. The ABRE element was found in 76.47% of *SlIDD* members. The ABRE element is involved in the abscisic acid response. Moreover, six types of stress response elements were detected in *SlIDD* promoters, which were involved in anaerobic induction, low-temperature response, anoxic-specific induction, defense and stress response, drought induction, and wound response. The ARE element was found in the promoter regions of all *SlIDD* genes except for *SlIDD4*. As well, 11 types of elements were found to regulate growth and development in *SlIDD* promoters, such as seed-specific regulation (RY-element), meristem expression (CAT-box), endosperm expression regulatory elements (GCN4_motif), and so on (Figure 5; Appendix A). Based on the predicted results, *SlIDD* genes may play a variety of functions in plant growth.

### 2.7. Expression Profile Analysis of SlIDD Genes in Different Tissues

The expression pattern of a gene can provide clues to its specific functional roles. In this study, we obtained the transcriptome data of tomato cultivar Heinz (*Solanum lycopersicum*, cv Heinz). The 17 *IDD* genes displayed tissue-specific expression, categorized into three classes (Figure 6; Appendix A). Class I included *SlIDD2*, *SlIDD7*, and *SlIDD11*, which were broadly expressed in various tissues. In contrast, most genes in Class III, particularly *SlIDD5*, *SlIDD8*, and *SlIDD14*, had low expression levels and could not even be detected in some tissues.

In addition, all 17 *SlIDD* genes showed tissue-specific or spatiotemporal expression patterns. For instance, the expression of *SlIDD7* gradually enhanced in six stages of fruit development, peaking at 10 days after the breaker stage (breaker + 10 stage). *SlIDD3*, *SlIDD9*, and *SlIDD4* were mainly expressed in 1–3 cm fruits. The expression of *SlIDD2* reached a peak in the fruit color-breaking stage and then decreased sharply. The expressions of *SlIDD6* and *SlIDD10* were higher in fruit in the green ripening stage compared with other tissues. The findings indicated a pivotal role for these genes in the progression of fruit development, hinting at their significance in the maturation process. Furthermore, some genes in Class III also had tissue-specific expression, although the expression level was very low. For example, *SlIDD5* and *SlIDD14* were expressed at detectable levels in roots but were hardly detected in other tissues.

To investigate tissue-specific expression differences of SlIDD genes, real-time fluorescence quantitative PCR measured their relative expression levels in various tissues of wild-type tomato (*Ailsa Craig*), including roots, stems, leaves, shoot apical meristem (SAM), cotyledons, hypocotyls, buds, fully open flowers, 1 cm green fruits, fruits at 5 days after breaker stage (breaker + 5), and seeds. (Figure 7). We examined the relative expression levels of *SlIDDs* in various groups and observed that the two homologous genes, *SlIDD13* and *SlIDD15,* clustered in Group A exhibited similar expression patterns, both of which were abundantly expressed in the hypocotyls. However, *SlIDD13* transcript levels were harder to detect compared to *SlIDD15*. These results indicated that *SlIDD15* might be the primary functional gene among the two paralogous genes. In addition, another pair of paralogous genes, *SlIDD16* and *SlIDD17,* also had similar expression patterns and had high expression levels in flowers. *SlIDD1*, *SlIDD7*, and *SlIDD9* are highly expressed in cotyledon and have similar expression patterns. In Group B, *SlIDD3* was specifically highly expressed in 1 cm green fruits, suggesting its potential role in regulating tomato fruit development. Among the 17 *SlIDD* genes, *SlIDD12* and *SlIDD6* exhibited relatively high expression in seeds, indicating their possible involvement in tomato seed development. Overall, the expression of these 17 *SlIDD* genes in multiple tissues was significantly different, indicating that these *SlIDDs* lead to more obvious functional differentiation in the evolutionary process.

### 2.8. Subcellular Localization of the SlIDD8 and SlIDD15 Proteins

Previous studies have revealed that IDD proteins function and act as transcription factors in the nucleus. To determine whether the tomato SlIDD8 and SlIDD15 proteins are nuclear-localized, their full-length CDS (excluding terminators) were fused with GFP and driven by the *35s* promoter. The *35s::SlIDD8-GFP* and *35s::SlIDD15-GFP* vectors were then transiently transformed into *Nicotiana benthamiana* leaves using the *Agrobacterium* strain EHA105. Observations of tobacco epidermal cells showed that the GFP signal of the SlIDD8-GFP and SlIDD15-GFP fusion proteins was specifically detected in the nucleus, while the *35s::GFP* control exhibited signals in both the cytoplasm and nucleus (Figure 8). This observation provides evidence that SlIDD8 and SlIDD15 are specifically localized and function in the nucleus.

### 2.9. CRISPR/Cas9-Mediated SlIDD15 Mutants Do Not Exhibit Defects in Shoot Gravitropism

The *Arabidopsis idd15*/*sgr5* mutant exhibited an obviously reduced gravitropic response in inflorescence stems, a phenotype that is significantly enhanced in the *idd14 15 16* triple mutant [24]. In addition, the *AtIDD15* ortholog *Loose Plant Architecture1* (*LPA1*/*OsIDD14*) also influences shoot gravitropism and architecture in rice [34], implying the conserved shoot gravitropic response function of *IDD15* in higher plants. To explore the function of *SlIDD15* in tomato, we first examined the expression pattern of *SlIDD13* and *SlIDD15* in Group A, which contained *AtIDD15* and *LPA1*. Results indicated that the expression level of *SlIDD15* was significantly higher than *SlIDD13* (Figure 7). Thus, we generated loss-of-function mutants of *SlIDD15* through CRISPR/Cas9-mediated genome editing technology. To enhance the editing efficiency, we selected two editing targets in the exon of the *SlIDD15* gene and integrated them into one construct (Figure 9a). Through PCR amplification and sequencing, two independent mutant lines were confirmed for further analyses (Figure 9b). *SlIDD15*-M1 harbored 1 bp deletion, and *SlIDD15*-M2 had a 5 bp deletion in the second exon, both of which resulted in fragment frameshift and premature translation termination (Figure 9c).

Next, the phenotype of these two independent mutant alleles was analyzed. The two allelic mutant plants were placed horizontally to observe the gravitropic response of stems. The gravitropic response of the two mutant plant stems was similar to the wild type, with no apparent defects (Figure 9d,e). This indicated that the single *SlIDD15* gene mutation is not sufficient to cause the gravitropic response defect of tomato stems, implying that there may be other factors that function redundantly with *SlIDD15*.

## 3. Discussion

In the past decades, IDD protein members have been gradually identified and reported across diverse species [3,40,41,42,43,44]. Research has found that IDD family proteins play an important role in transcriptional regulation during plant development and in response to environmental stress [6]. However, the systematic identification and function annotation in the horticultural plant tomato have been rarely reported. In this study, we conducted a comprehensive genome-wide identification in tomato and found 17 *SlIDD* genes distributed on ten chromosomes, excluding Chr 5 and Chr 12 (Figure 1). These genes encode IDD proteins characterized by the ID domain, consistent with the typical structure found in IDD proteins across different plant species [3,40]. Within the reported C2H2 zinc finger protein family of tomato, the IDD gene family has been phylogenetically assigned to a distinct clade, comprising a total of 18 members, which is consistent with the identification of the tomato *IDD* gene family reported recently [45,46]. Only 17 members are contained in this study. The *Solyc05g054030.3.1* member with C2HR was excluded, and only members of the typical ID domain were retained. Diploid species exhibit uniform member counts in the *IDD* gene family, whereas polyploid species, exemplified by bamboo and *Brassica napus* with 32 and 58 members, respectively, possess a higher number of *IDD* gene family members [40,42]. The investigation of conserved motifs and gene structure demonstrated that the SlIDD proteins contained ten conserved motifs, and the 17 *SlIDD* genes exhibited a consistent three to four exons (Table 1; Figure 2). Furthermore, it was predicted that SlIDD proteins are localized to the nucleus., and transient expression of SlIDD8 and SlIDD15 proteins in tobacco confirmed the prediction (Table 1; Figure 8), which is consistent with previous reports [10]. This means that the *IDD* genes have remained relatively conserved throughout evolutionary processes.

In the evolutionary analysis of C2H2 proteins in tomato, the IDD proteins were found to cluster separately into a distinct branch, differentiating them from other C2H2 zinc finger proteins. This observation suggests that the IDD protein family is highly stable and conserved in evolution [45]. IDD protein evolution in angiosperms could be divided into eight monophyletic lineages [47]. In this study, the phylogenetic analysis categorized 71 IDD proteins into seven subgroups based on the previously reported topology of our phylogenetic tree [47]. Each group contained an SlIDD protein (Figure 3). In Group B, there were only members from tomato and *Arabidopsis*, suggesting that this clade might be specific to dicots (Figure 3). Meanwhile, Group C lacked AtIDD protein members, indicating that despite tomato and *Arabidopsis* being more closely related evolutionarily, the tomato proteins in this clade are more closely related to those in monocots (Figure 3). Gene duplication and evolutionary analysis suggested that *SlIDD* genes may undergo expansion of family members during evolution, with fragment replication being the primary mechanism of this evolution rather than tandem duplication (Figure 4). The expansion of gene families is influenced by environmental pressure and selection pressure. The calculated Ka/Ks ratio was less than one (Table 2), indicating that the *SlIDD* gene family experienced strong purification selection pressure. In the collinearity analysis between tomato and other species, we found that *SlIDD* had more collinear pairs with dicotyledons than with monocotyledons (Figure 4). The results indicate that tomatoes have a closer evolutionary relationship with dicotyledonous plants. *SlIDD1*, *SlIDD2*, and *SlIDD13* have collinear gene pairs in all four species, which suggests that they may have remained relatively stable throughout long-term evolution. In contrast, other members may have undergone more evolutionary events such as mutations, losses, or rearrangements during evolution, resulting in lower conservation among species.

Analyzing *cis*-acting elements in genes and expression patterns, we can infer how these elements affect gene expression, revealing their potential role in organism function, development, or response to the environment. All *SlIDD* gene promoters contained light-responsive elements, indicating that *SlIDD* genes can be induced by light. *IDD* genes are related to hormones and stress response, which is similar to the *cis*-acting elements of IDD protein in other species, suggesting that *IDD* gene function may also be conserved [39,41,43]. This suggests that tomato and related *IDD* genes are actively involved in regulating plant growth, development, and responses to environmental stress. Functional characterization of some IDD proteins has shown that they play crucial regulatory roles in seed development, root development, flowering transition, and hormone signaling in *Arabidopsis* and rice [6]. It has been reported that overexpression of *AtIDD1/ENY* disrupt seed development, delaying endosperm depletion and testa senescence, leading to a shortened maturation program [25]. Distinctly, *AtIDD1* promotes germination by counteracting aspects of seed maturation associated with abscisic acid [25]. Furthermore, *GAF1* (*AtIDD2*, *CARRION CROW*) exhibits a dual role in modulating germination, serving as either an inhibitor or a stimulant, contingent on the availability of gibberellic acid (GA) [48]. In Group E of the phylogenetic tree, *SlIDD6, SlIDD16, and SlIDD17 were* grouped together with *AtIDD1* and *AtIDD2*. The qRT-PCR analysis showed that *SlIDD6* was highly and specifically expressed in tomato seeds, while *SlIDD16* and *SlIDD17* also exhibited relatively high expression levels in seeds. In addition, *ZmIDDveg9* (*NKD1*) and *ZmIDD9* (*NKD2*) play a role in regulating seed maturation in maize [26,27]. Phylogenetic analysis showed that *SlIDD12* was close to *ZmIDDveg9* (*NKD1*) and *ZmIDD9* (*NKD2*) in the same clade Group D, and *SlIDD12* was highly expressed in seeds (Figure 7). The findings imply that *SlIDD12* may contribute significantly to the process of seed maturation and development. We observed that *SlIDD14* clustered with *AtIDD3* and *AtIDD8* in Group F. Notably, *SlIDD14* exhibited specifically high expression levels in roots. Previous studies have shown that both *AtIDD3* and *AtIDD8* are involved in root development. This suggests that *SlIDD14* may have similar functions in root development to the *AtIDD3* and *AtIDD8*. Interestingly, the collinear gene pairs of *SlIDD* exhibited similar expression patterns in various tissues, like *SlIDD16* and *SlIDD17*, *SlIDD7* and *SlIDD9*, and perhaps they perform similar functions. Even if expression pattern analysis can suggest potential functions, the function of these genes needs to be further verified.

Tomato is a globally important horticultural crop, and exploring their fruit development and ripening is of great significance. During the development and ripening process of tomato fruits, a series of plant hormones, transcription factors, and epigenetic modifiers have been reported to play crucial roles. Plant hormones such as ethylene and gibberellin [49], transcription factors such as the vital genes *NOR*, *RIN*, *CNR*, and *LOB* [50], and epigenetic modifiers such as the histone demethylase *SlJMJ6* [51], providing a foundational understanding of the molecular regulatory network governing fruit development and ripening process. As identified by ChIP-chip and transcriptome analysis, *SlIDD1* is one of the transcription factors (TFs) that are directly targeted and positively regulated by RIN [52]. The qRT-PCR also showed that the expression levels of *SlIDD1* decreased in *rin*, indicating its potential role in fruit maturation [52,53]. Transcriptome profiling of tomato fruit development shows that SlIDD1 and SlIDD4 are related to the biosynthesis of ascorbic acid, carotenoids, and flavonoids [54]. Repression of *SlLOB1* in transgenic fruit delays softening, whereas its overexpression throughout the plant via the 35S promoter accelerates cell wall gene expression and causes premature softening [55]. Based on GO term enrichment, *SlIDD9* displayed substantial downregulation in both the locular gel and pericarp in fruit from both *SlLOB1*-repressed lines [55]. Through a weighted gene co-expression network analysis (WGCNA) of tomato fruits, *SlIDD10* was found to be associated with the accumulation of ascorbate and phenolics [56].

The nuclear-localized transcriptional activator *MaC2H2-IDD* exerts a pivotal role in fruit ripening, as evidenced by its transient and ectopic overexpression accelerating the ripening process in both “Fenjiao” banana and tomato, whereas its transient silencing inhibits the ripening of “Fenjiao” banana fruits [57]. The evolutionary tree indicates that the identified *MaC2H2-IDD* is a homologous gene with *SlIDD5* [57]. Our study demonstrated that *SlIDD5* was specifically highly expressed in 1 cm green fruits (Figure 7), suggesting that this gene may play a similar role in fruit maturation. However, there are fewer reports on the role of *IDD* family members in fruit development and ripening. In this study, it was found that some *SlIDD* members were expressed at different stages of fruit development. Specifically, *SlIDD7* expression gradually increased over six stages of fruit development, reaching its highest level at 10 days after the breaker stage. *SlIDD3*, *SlIDD9*, and *SlIDD4* were mainly expressed in 1–3 cm fruits. *SlIDD2* expression peaked at the fruit color-breaking stage and then sharply decreased. The expressions of *SlIDD6* and *SlIDD10* were higher in the green ripening stage compared with other tissues. Future research will involve verifying the genetic and regulatory relationships among these *SlIDD* members and the key factors already reported to participate in tomato fruit development and ripening processes. This will elucidate new mechanisms by which *SlIDD* controls fruit development and ripening, providing new insights to assist horticultural crop breeding.

Gravitropism affects plant architecture, thereby influencing the plant reproductive traits [29]. Thus, the gravitropism-induced plant architecture has been taken into consideration for crop genetic improvement. During the gravitropic response process in plants, signal factors perceive gravity signals, leading to the asymmetric distribution of auxin, which, in turn, causes gravitropic growth. *AtIDD15* and its rice functional ortholog *LPA1* both belong to the Group A clade and can regulate gravity sensing, implying that the tomato proteins in this clade are likely to conservatively regulate gravity sensing. This study also found differences among species. The loss-of-function mutations of *AtIDD15* in *Arabidopsis* or *LPA1* in rice significantly affect plant gravity sensing [34,58], indicating that in both *Arabidopsis* and rice, gravity sensing is primarily regulated by a single major gene. However, in tomato, the Group A clade had *SlIDD13* and *SlIDD15,* two members; although *SlIDD15* was expressed at significantly higher levels, its loss-of-function mutation generated by CRISPR-Cas9 did not result in a noticeable gravitropic response phenotype (Figure 9). In our experiments, we obtained a double-mutant plant with mutations in both *SlIDD15* and *SlIDD13* but only a single individual. Gravitropism assays conducted on this double-mutant plant showed a weakened shoot gravitropism phenotype (Appendix A). This result suggests that, unlike the single major gene functions in *Arabidopsis* and rice, it is likely that *SlIDD13* and *SlIDD15* redundantly function in regulating gravity sensing. Thus, future genetic studies are needed to verify the phenotypes and functions of *SlIDD13* and *SlIDD15* double mutants, which will aid in understanding their mechanisms and contribute to tomato plant architecture breeding.

## 4. Materials and Methods

### 4.1. Genome-Wide Identification and Physicochemical Properties Analysis of IDD Gene in Tomato

BLAST retrieval and domain identification were used to obtain candidate *SlIDD* members. The protein sequences for the 16 *AtIDD* genes, as documented in previous studies, were accessed from the TAIR database (https://www.arabidopsis.org/, accessed on 1 December 2023) [5]. Meanwhile, the protein sequences for tomato, based on the ITAG 4.0 annotation, were sourced from the Sol Genomics Network, a comprehensive tomato genome database [59] (https://solgenomics.net/projects/tomatodisease/, accessed on 1 December 2023). Next, 16 AtIDD protein sequences were utilized as queries to search tomato protein databases using local BLAST of TBtools [60]. The screening parameter E-value was set to be less than 1 × e^−10^, and the redundant sequences with the same gene ID were deleted to obtain the preliminary screened IDD candidate members of tomato. Then, we downloaded the Pfam-A.hmm file from the Pfam database (http://pfam.xfam.org/, accessed on 1 December 2023). Using the simple HMM search function in TBtools, the tomato protein sequences containing zf-C2H2_jaz (PF12171) and zf-C2H2_6 (PF19312) domains were further retrieved [40], and members with E values less than 1 × e^−10^ were identified as candidates. Subsequently, we submitted the candidate protein sequences, jointly isolated by BLAST and domain, to InterPro (http://www.ebi.ac.uk/interpro/, accessed on 27 February 2024) to further verify whether they belonged to the IPR031140 (a family of plant-specific transcription factors, including protein indeterminate-domain 1–16 from *Arabidopsis* containing the conserved INDETERMINATE DOMAIN with the four zinc finger motifs gene family [61]. After removing members that did not belong to the IPR031140 gene family, MEGA11 was applied to the sequence alignment of the obtained candidate protein sequences and manually checked for ID domain [62]. The protein sequences lacking complete ID-domain were discarded, and only the domains with typical two C2H2 and two C2HC zinc-finger structures were retained. Finally, we renamed the obtained candidate members as SlIDD1–SlIDD17 based on their protein sequence similarity with ZmID1, which was calculated using TBtools.

In addition, we used tomato genome data and annotation files to extract genome locations of the corresponding 17 *SlIDD* genes using TBtools. Subsequently, we harnessed the Expasy platform (http://web.expasy.org/protparam/, accessed on 27 February 2024) to extensively analyze the physicochemical characteristics of SlIDD proteins, encompassing the amino acid number, molecular mass, predicted isoelectric point, instability parameter, aliphatic index, and hydrophilic grand average of hydropathicity, among other pertinent properties [63]. The subcellular localization of SlIDD proteins was predicted by CELLO (http://cello.life.nctu.edu.tw/, accessed on 8 March 2024) online website [64].

### 4.2. Multiple Sequence Alignment and Chromosomal Localization Analysis

We used DNAMAN (version 8) software to conduct dynamic multiple sequence alignment of SlIDDs and ZmID1 proteins and mapped conserved N-terminal and C-terminal domains. Then we utilized the extensive GFF3 annotation data for the tomato reference genome. Employing the advanced visualization features of TBtools, we constructed a detailed chromosome map, accurately pinpointing the locations of *SlIDD* genes on the chromosomes.

### 4.3. Phylogenetic Relationship, Gene Structure, and Conserved Motif Analysis of SlIDDs

We used the ClustalW plugin in MEGA11 to perform multiple sequence alignment of SlIDD protein full-length sequences and generated the alignment file, which was used to construct a neighbor-joining evolutionary tree. To predict conserved motifs, we submitted the SlIDD protein sequences to the MEME (https://meme-suite.org/meme/tools/meme, accessed on 28 February 2024) website, and the parameters were set to ten motifs (motif E-value less than 0.05) [65]. Gene structure details for *SlIDDs* were extracted from the GFF3 annotation file of the tomato genome. Finally, TBtools was employed to visualize the evolutionary tree, conserved motifs, and gene structures [60].

### 4.4. Phylogenetic Analysis of IDDs

The reported IDD protein sequences of *A. thaliana*, *O. sativa,* and *Z. mays* were downloaded from Phytozome (http://www.phytozome.org, accessed on 19 October 2023) [66]. Details of the *IDD* gene accession numbers for these three species are shown in Appendix A. The SlIDD protein sequences were extracted from the tomato ITAG 4.0 protein sequence file by TBtools. Then, multiple sequence alignment of 71 IDD protein sequences was performed by the MEGA11 software 11. Then an evolutionary tree was constructed using the neighbor-joining method [67]. The ChiPlot online tool was used to optimize the phylogenetic tree to present the evolutionary tree more clearly. (https://www.chiplot.online/, accessed on 8 March 2024).

### 4.5. Gene Duplication, Collinearity Analysis, and Ka/Ks Calculation

Genome files of *Arabidopsis* (*Athaliana*_447_Araport11), rice (*Osativa*_204_v7.0), maize (*Zmays*_493_RefGen_V4), tomato (*Slycopersicum*_691_ITAG4.0), and potato (*Stuberosum*_686_v6.1) were retrieved from the Photozome database (http://www.phytozome.org, accessed on 7 March 2024) [66]. Using the One Step MCScanX plugin in TBtools, we analyzed *SlIDD* gene duplication events and their collinear relationships with genes from other species, following the default parameter settings. The Advanced Circos program of TBtools was used to generate a chromosome collinearity map of *SlIDDs*, and the dual synteny plotting tool of TBtools was employed to visualize and map the collinear gene pairs of tomato and other species [60,68]. By employing the Simple Ka/Ks Calculator module integrated within TBtools, we calculated the rates of both non-synonymous (Ka) and synonymous (Ks) mutations specifically for the collinear gene pairs that had been previously identified. The duplication time (T) was calculated using the formula T = Ks/2λ × 10^−6^ million years ago (Mya) (approximate value for clock-like rate λ = 1.5 × 10^−8^ years) [69,70].

### 4.6. Analysis of the Cis-Acting Elements in the Promoters of SlIDDs

According to the genomic annotation information of tomato, TBtools was used to extract putative promoter sequences of *SlIDD* genes with a coding sequence upstream length of 2 kb. Subsequently, the promoter sequences underwent prediction and analysis by utilizing the PlantCARE online platform (http://bioinformatics.psb.ugent.be/webtools/plantcare/html/, accessed on 8 March 2024) [71]. After classification and statistical analysis, TBtools was used to make a heat map of promoter *cis*-acting elements [60], and the Chiplot online platform (https://www.chiplot.online/, accessed on 8 March 2024) was used to make classified statistical bar charts.

### 4.7. The Expression Profiles of SlIDD Genes in Different Tissues

In order to explore the expression profile of tomato *SlIDD* genes, RNA-seq data were acquired from the Tomato Functional Genomic Database (http://ted.bti.cornell.edu/cgi-bin/TFGD/digital/home.cgi, accessed on 25 March 2024) under accession number D004 [72]. FPKM (fragment per million exons mapping) data of *Solanum lycopersicum* cv. Heinz 1706 in different tissues and developmental stages were retrieved. Then TBtools was used to construct an expression heat map with the clustering [60].

### 4.8. RNA Extraction, Reverse Transcription, and Quantitative Real-Time RT-PCR (qRT-PCR) Analysis

Different tissues of tomato (*Ailsa Craig*) were collected for RNA extraction. For tomato fruit tissues, the TransZol Plant kit (TransGen Biotech, ET121-01, Beijing, China) was utilized to extract total RNA, while for other tissues, the RnaEx™ Total RNA isolation kit (GENEray Biotech, Shanghai, China) was employed to isolate the total RNA content. The quality and concentration of RNA were determined with the NanoDrop 2000 spectrophotometer (Thermo Fisher Scientific, Shanghai, China) and 2% agarose gel electrophoresis. Subsequently, about 3 μg of total RNA were reverse transcribed with HiScript^®^ II 1st Strand cDNA Synthesis Kit (Vazyme, Nanjing, China) for quantitative PCR. The specific primers of *SlIDD* genes were designed on the NCBI (http://www.ncbi.nlm.nih.gov/tools/primer-blast, accessed on 14 March 2024) website [73], and the tomato *Actin* gene (*Solyc03g078400*) was used as the internal reference. Primer sequence information is in Appendix A. We utilized the QuantFast SYBR Green qPCR kit from Magic-bio (Hangzhou, China) to conduct quantitative real-time PCR analyses on a LightCycler 480 platform (Roche, Basel, Switzerland). The quantification and assessment of the expression levels of targeted genes were performed by employing the 2^−∆∆Ct^ method [74]. The qRT-PCR experiment was repeated in three biological and technical replicates. Finally, GraphPad Prism 8 was used to draw the graph of relative expression.

### 4.9. Subcellular Vector Construction and Subcellular Localization

Using specific primers for SlIDD8 and SlIDD15, PCR amplification was performed with Ailsa Craig cDNA as the template. The PCR products were purified, and the vector was digested with *XhoI* and *KpnI*. Subsequently, the purified PCR products and digested vector were ligated using a seamless cloning enzyme. The ligation products were then transformed into competent *E. coli* DH5α. Colonies with correct bands were selected for plasmid extraction and subsequently transformed into *Agrobacterium tumefaciens* strain EHA105. Multiple monoclones were selected and identified, and the positive clones were cultured at 28 °C and expanded. Then the cultures were centrifuged, resuspended in an infiltration solution (10 mM MgCl_2_, 10 mM MES, 100 µM acetosyringone, pH 5.6), and adjusted to an optical density of OD600 = 1.0. For tobacco transformation, four-week-old robust *Nicotiana benthamiana* plants were used. The mixed suspension was injected into healthy tobacco leaves, and the infected tobacco was cultured in dark conditions for two days. The distribution of the signal in the tobacco leaf cells was then observed using a confocal microscope.

### 4.10. CRISPR/Cas9-Mediated Gene Editing in Tomato

The CRISPR/Cas9 editing targets for *SlIDD15* were designed using the CRISPR-GE tool (http://skl.scau.edu.cn/, accessed on 28 March 2023) [75]. Target sequences were inserted into two sgRNA expression cassettes via overlapping PCR, utilizing specific primers for each sgRNA (Appendix A). The first round PCR was conducted with primers U-F, *SlIDD15*-AtU3d-R T1 (or *SlIDD15*-AtU3b-R T2), *SlIDD15*-AtU3d-F T1 (or *SlIDD15*-AtU3d-R T2), and gR-R (Appendix A). The secondary PCR was performed using site-specific primer pairs (Pps-GAL/Pgs-GA2 for Target 1 and Pps-GA2/Pgs-GAR for Target 2), incorporating *BsaI* restriction sites (Appendix A). Finally, the sgRNA cassettes were then ligated into the *pYLCRISPR/Cas9P35s-N* vector via Golden Gate ligation [76]. The confirmed *pYLCRISPR/Cas9P_35s_-N-SlIDD15* binary vector was transferred into *Agrobacterium tumefaciens* strain EHA105. Transgenic plants were generated through the Agrobacterium-mediated cotyledon transformation method described by Van et al. [36]. Genomic DNA was isolated from the transgenic plants for PCR amplification and sequencing to validate the mutation status. The homozygous T_1_ plants of *SlIDD15*-M1 and *SlIDD15*-M2 were chosen to perform phenotype analyses. Detailed primer sequences used are shown in Appendix A. All primer syntheses were completed at GENEray Biotechnology (GENEray, Shanghai, China), and sequencing was completed at Songon Biotechnology (Sangon, Shanghai, China).

### 4.11. Gravitropism Assay

To examine the gravitropic responses of tomato stems, intact tomato plants with a height of 30 to 50 cm were used. The gravistimulation was given by rotating the plates 90° in darkness at 23 °C. The curvature of the stem was assessed by measuring the angle between the direction of apex growth and the horizontal baseline. At least three individuals of each genotype were examined, and the bending angle was calculated using Image J software.

### 4.12. Plant Materials and Growth Condition

The plants were grown in greenhouses. The conditions were set to control the temperature at 18~22 °C, the indoor humidity was kept between 60% and 70%, and the lighting conditions were set to 15 h of daylight and 9 h of darkness cycle for day and night. The growth medium consisted of a mixed soil formula in the ratio of 3 parts coco peat, 1 part humus soil, 1 part perlite, and 1 part vermiculite. Alternating irrigation with both clear water and water-soluble fertilizer was implemented, and pest control measures were taken by applying pesticides for disease and insect prevention twice weekly.

## 5. Conclusions

In general, we took the *SlIDD* gene family as the research object, and the protein sequences of the *SlIDD* gene family members were analyzed by bioinformatics, including their physicochemical properties and conserved domain analysis. Additionally, the evolutionary relationship, collinear relationship, and *cis*-acting elements of the family were predicted. Moreover, qRT-PCR assays verified that the *SlIDD* genes are expressed in different tissues. Subcellular localization demonstrated that the SlIDD15 and SlIDD8 proteins are localized in the nucleus. Furthermore, we selected *SlIDD15* in Group A to carry out functional research on CRISPR/Cas9 gene editing technology. The CRISPR-edited *SlIDD15* mutant plants showed normal stem gravitropism, implying the existence of other factors that are redundant with *SlIDD15* functionality. In the future, it will be necessary to clarify the gene functions of various *IDD* members in tomato, especially the molecular regulatory network of *IDD15* in the process of gravitropism and tomato plant architecture, thus providing a theoretical basis for improving the above-ground plant architecture of tomato. This study provides a theoretical basis and lays the foundation for future research on the function and mechanism of action of *SlIDD* genes in plant growth and development.

## Figures and Tables

**Figure 1 ijms-25-10422-f001:**
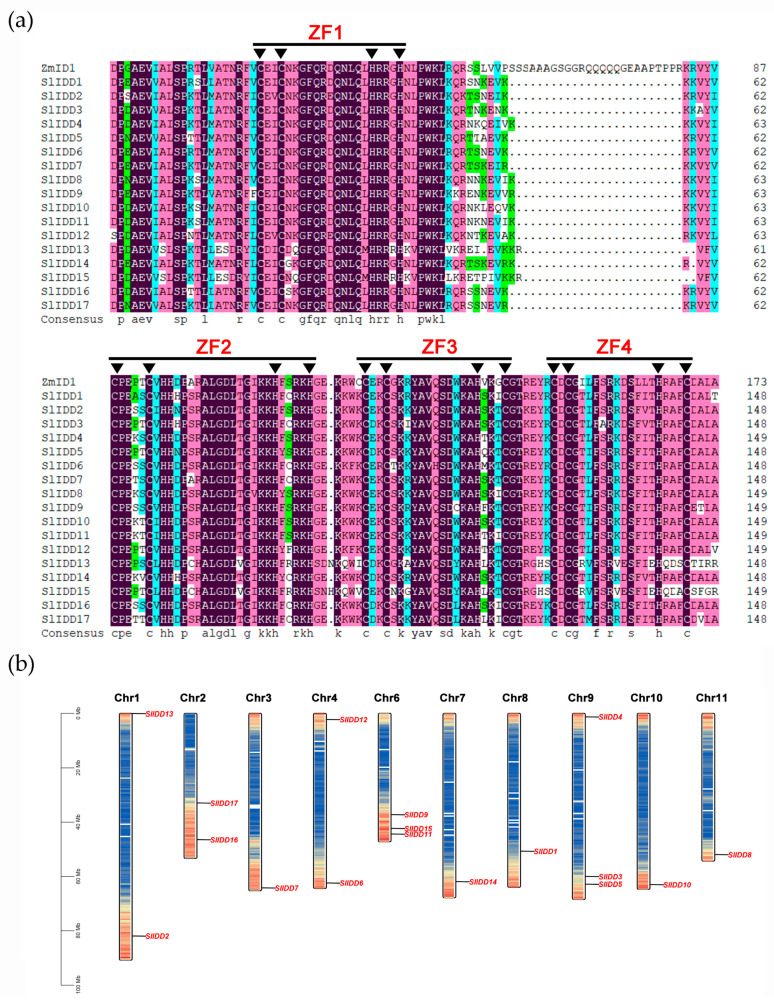
Alignment of SlIDD protein sequences and chromosome localization. (**a**) The N-terminal conserved ID-domain of the 17 SlIDD and ZmID1 protein sequences was aligned by DNAMAN. Zinc finger domains are highlighted with a straight line. Conserved amino acids are marked with inverted triangles, showing CCHH in ZF1 and ZF2 and CCHC in ZF3 and ZF4. Amino acids that were identical are displayed in text on a black background. (**b**) Chromosome names are listed at the top of each chromosome. Gene locations are indicated by black lines. Within chromosomal segments, blue lines mark zones characterized by low gene density, whereas red lines distinguish areas featuring high gene density. A chromosome length scale is marked on the left side of the figure.

**Figure 2 ijms-25-10422-f002:**
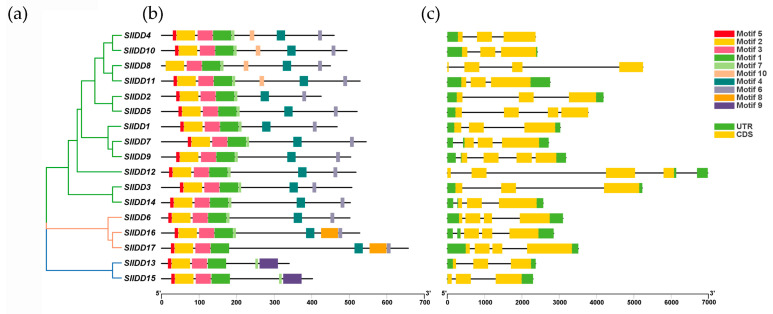
Evolutionary analysis, motifs, and gene structures of the *SlIDD* gene family in tomato. (**a**) Neighbor-joining (NJ) phylogenetic tree of SlIDDs generated with MEGA11 using 1000 bootstrap replicates. Subgroups are highlighted with colored lines. (**b**) Conserved motif profiles of SlIDDs. Ten distinct motifs are represented by differently colored boxes, and the scale represents 100 amino acids. Motif sequences are shown in Appendix A. (**c**) Analysis of exon–intron structure in *SlIDD* genes. The black lines represent introns. The yellow and green squares indicate the coding sequence (CDS) and UTR, respectively. The scale bar represents 1 kb.

**Figure 3 ijms-25-10422-f003:**
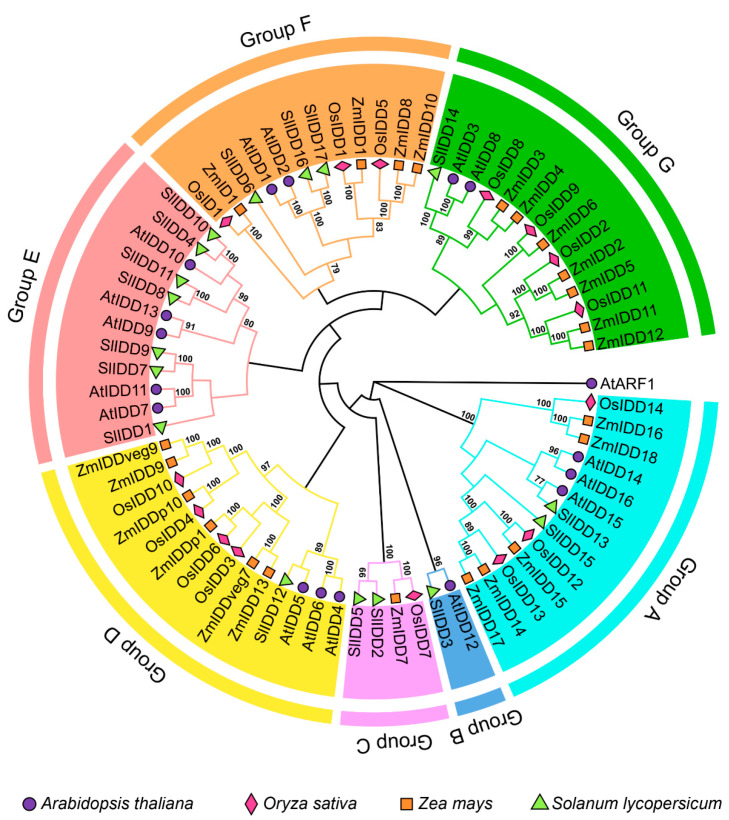
Phylogenetic relationship analysis of IDD proteins from *S. lycopersicum* (Sl), *A. thaliana* (At), *Z. mays* (Zm), and *O. sativa* (Os). A phylogenetic neighbor-joining tree was generated with 1000 bootstrap replicates in MEGA11. Different colors of outer rings are used to distinctly differentiate the groups in the evolutionary tree. Distinctive symbols are used to differentiate IDD proteins from different species, with purple solid circles marking *A. thaliana*, pink diamonds marking *O. sativa*, orange squares marking *Z. mays*, and green triangles marking *S. lycopersicum*. Only display nodes with bootloader support values greater than 60 are presented.

**Figure 4 ijms-25-10422-f004:**
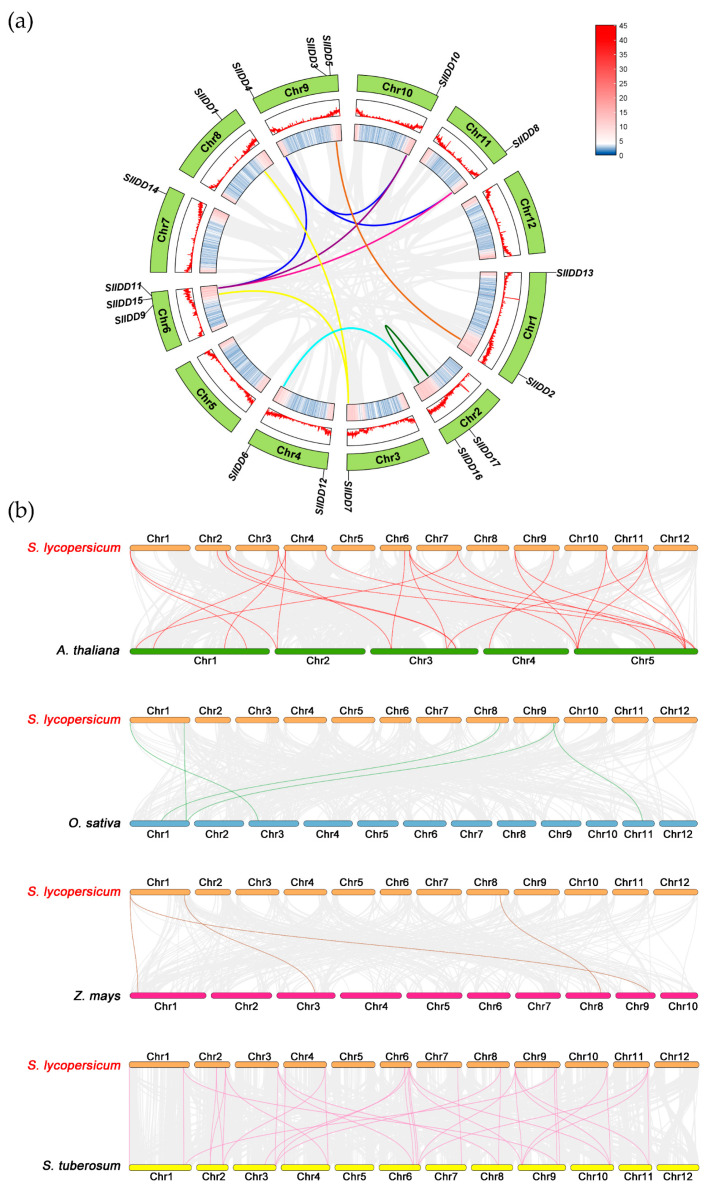
The collinearity analysis of *SlIDD* family members. (**a**) Intraspecies collinearity of *SlIDD* genes in tomato, with gray lines representing all collinear blocks and colored curves linking segmentally duplicated *SlIDD* genes. The two inner rings show the gene density on the chromosome by heat map and line map, respectively. (**b**) Interspecies collinearity analysis of *IDD* genes between tomato and four other plants. Gray lines indicate collinearity between tomato and *Arabidopsis*, as well as between rice, maize, and potato. The red, green, brown, and pink lines are used to highlight the *IDD* gene pairs between tomato and other species, respectively.

**Figure 5 ijms-25-10422-f005:**
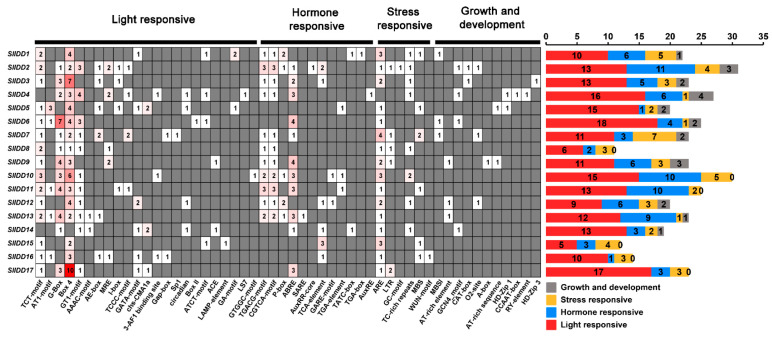
The identification of *cis*-acting elements in the promoters of *SlIDD* genes. The grid in the heatmap is color-coded from white to red in a gradual transition, marking the quantity of *cis*-acting elements. The values displayed in the grid represent the number of these *cis*-acting elements. The stacked graph represents the count of four categories of *cis*-regulatory elements statistically analyzed in the *SlIDD* genes.

**Figure 6 ijms-25-10422-f006:**
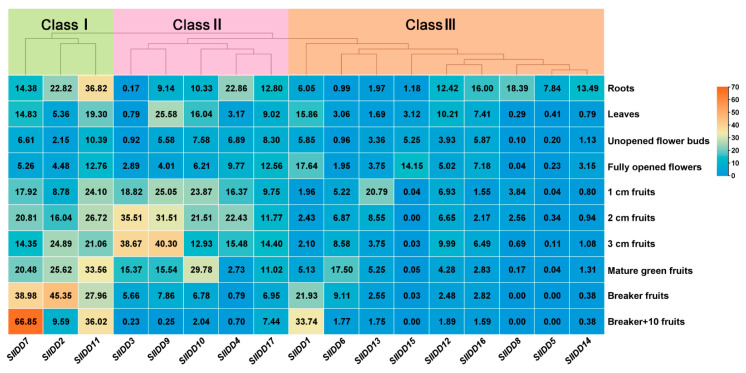
Heatmap of *SlIDD* expressions in a variety of tissues. All data were acquired from the TFDG database. The values represent RPKM (reads per kilobase per million mapped reads). This dataset consists of Illumina RNA-seq analysis of leaves, roots, flower buds, fully opened flowers, and 1 cm, 2 cm, 3 cm, mature green, breaker, and breaker + 10 fruits of tomato cultivar Heinz.

**Figure 7 ijms-25-10422-f007:**
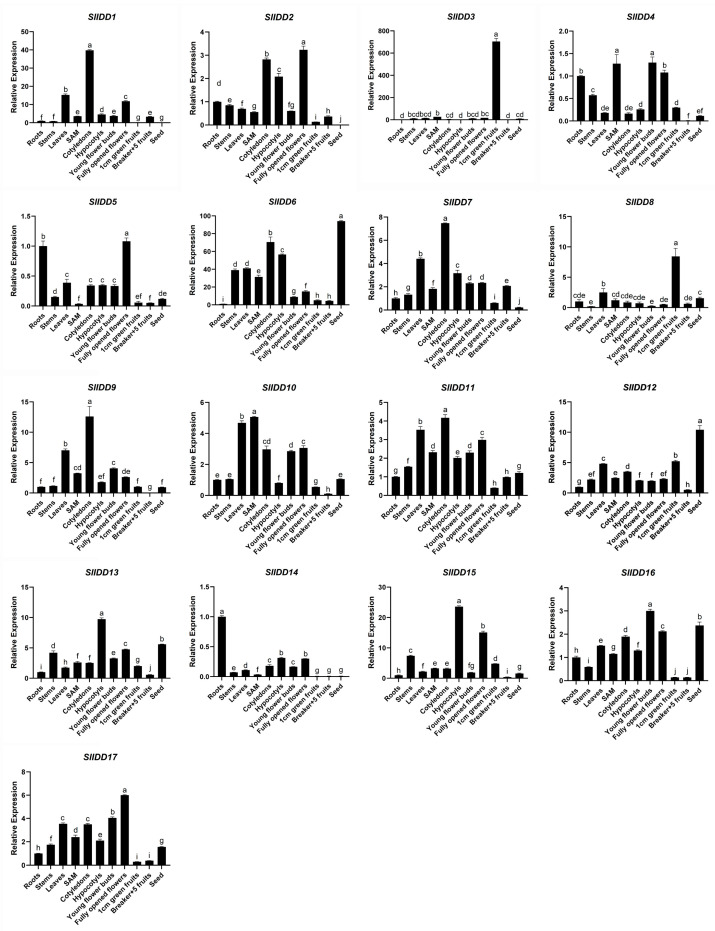
Expression pattern of *SlIDDs* of *Ailsa Craig*. The vertical axis represents relative expression levels, while the horizontal axis represents different tissues and the same tissue at different developmental stages. The error bars in the figure represent standard deviations (SD) (n = 3). Data represent the mean of three biological experiments ± standard error of the mean. The root relative expression values were normalized to 1. The significance analysis was performed using a one-way ANOVA test, and the significance level was *p* < 0.05. According to Duncan’s multiple range tests, the presence of different lowercase letters indicates statistically significant differences among the groups at a significance level of *p* < 0.05.

**Figure 8 ijms-25-10422-f008:**
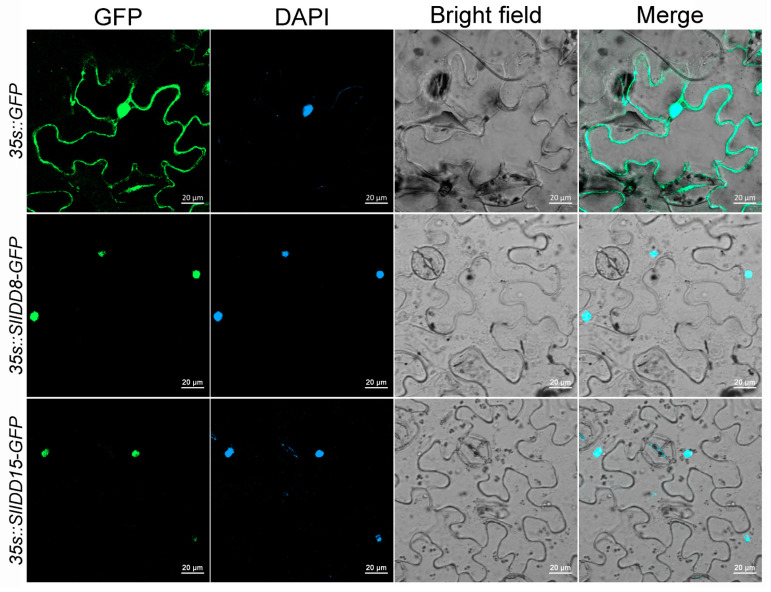
Subcellular localization of SlIDD15 and SlIDD8 protein. Images displayed from left to right show green fluorescent protein (GFP), DAPI (fluorescent dye for staining cell nuclei), bright field, and an overlay (GFP with bright field) from the same sample. The *35s::GFP* fusion served as the positive control for the protein.

**Figure 9 ijms-25-10422-f009:**
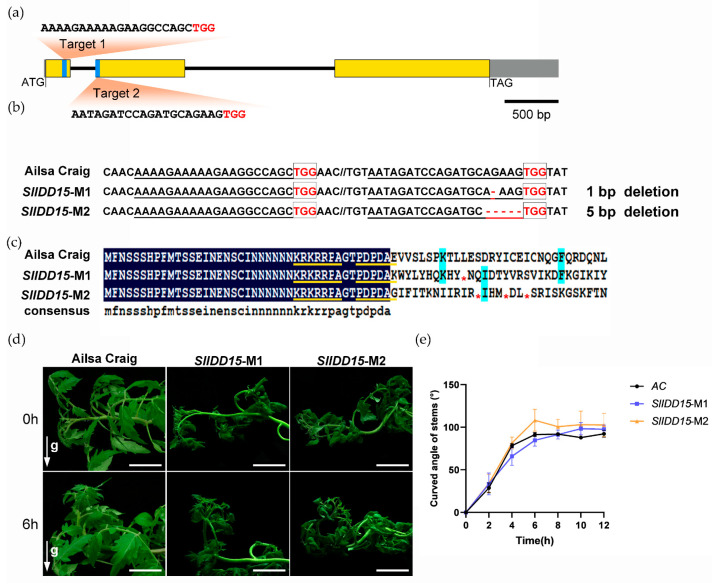
CRISPR/Cas9-mediated *SlIDD15* gene mutations and shoot gravitropism phenotypic analysis. (**a**) Schematic diagram of the sgRNA target sites on the *SlIDD15* gene. The exons are indicated by yellow boxes, introns are indicated by straight lines, and the UTR are indicated by grey boxes. The PAM motifs are marked in red. Bar = 500 bp. (**b**) Schematic diagram of multiple sequence alignment between homozygous mutant lines and wild type *Ailsa Craig.* PAM motifs are highlighted with boxes, target sites are underlined, and dashes represent deletions. The red stars indicates termination of protein coding (**c**) Amino acid sequence alignment of *SlIDD15*-Ms proteins. Protein sequence alignment was performed utilizing the DNAMAN 6 software. The target site has been underlined in yellow. (**d**) Phenotypic analysis of gravitropic response in the T_1_ gene-edited mutant lines, *SlIDD15*-M1 and *SlIDD15*-M2. The direction of the gravity vector is indicated by the white arrow g. Bar = 5 cm. (**e**) Statistical analysis of the gravitropic response of *SlIDD15*-Ms stems after altering the direction of gravity (n = 3).

**Table 1 ijms-25-10422-t001:** Identification and analysis of the physicochemical properties of IDD proteins in tomato.

Gene Name	Gene ID	Identity with ZmID1 (%)	Number of Amino Acid	Molecular Weight	Theoretical pI	Instability Index	Aliphatic Index	Grand Average of Hydropathicity	Number of Exons	Subcellular Localization
*SlIDD1*	*Solyc08g063040.4.1*	44.95	467	51,814.78	9.17	45.49	53.55	−0.774	3	Nuclear
*SlIDD2*	*Solyc01g099340.3.1*	44.66	424	47,449.69	8.92	49.55	58.68	−0.898	3	Nuclear
*SlIDD3*	*Solyc09g065670.3.1*	44.41	506	55,734.15	8.7	39.66	60.02	−0.679	3	Nuclear
*SlIDD4*	*Solyc09g007550.3.1*	42.45	459	51,314.57	9.38	46.92	58.47	−0.775	3	Nuclear
*SlIDD5*	*Solyc09g074780.3.1*	40.33	520	57,535.07	9.06	54.7	60.25	−0.771	4	Nuclear
*SlIDD6*	*Solyc04g080130.3.1*	40.2	501	54,292.99	8.24	50.04	67.01	−0.477	4	Nuclear
*SlIDD7*	*Solyc03g121660.3.1*	39.17	544	59,497.5	8.93	47.93	56.01	−0.695	3	Nuclear
*SlIDD8*	*Solyc11g069240.2.1*	38.66	449	50,087.32	9.31	45.76	66.06	−0.72	4	Nuclear
*SlIDD9*	*Solyc06g062670.3.1*	38.62	503	56,295.12	8.75	51.69	61.91	−0.67	4	Nuclear
*SlIDD10*	*Solyc10g084180.2.1*	38.26	493	54,589.11	9.2	46.13	58.03	−0.686	3	Nuclear
*SlIDD11*	*Solyc06g075250.3.1*	37.96	528	58,184.82	9.09	41.29	58.07	−0.793	3	Nuclear
*SlIDD12*	*Solyc04g008500.4.1*	37.58	517	57,519.53	9.13	44.62	56.25	−0.855	4	Nuclear
*SlIDD13*	*Solyc01g005060.3.1*	36.92	339	38,602.66	8.77	79.82	69.38	−0.824	3	Nuclear
*SlIDD14*	*Solyc07g053570.4.1*	35.76	502	55,280.78	9.08	54.53	66.87	−0.651	3	Nuclear
*SlIDD15*	*Solyc06g072360.3.1*	35.16	401	45,860.71	9.12	67.09	66.88	−0.827	3	Nuclear
*SlIDD16*	*Solyc02g085580.4.1*	30.73	483	50,520.04	8.87	59.45	63.54	−0.351	4	Nuclear
*SlIDD17*	*Solyc02g062940.3.1*	28.02	656	68,007.62	8.31	56.09	60.87	−0.34	4	Nuclear

**Table 2 ijms-25-10422-t002:** Evolutionary parameters for duplicated *SlIDD* genes.

Duplicate Gene Pair	Ka	Ks	Ka/Ks	Purify Selection	Duplication Type	Time = Ks/2λ (MYAa)
*SlIDD7*/*SlIDD1*	0.327	1.710	0.191	Yes	Segmental	57.015
*SlIDD7*/*SlIDD9*	0.238	1.005	0.237	Yes	Segmental	33.515
*SlIDD2*/*SlIDD5*	0.337	2.803	0.120	Yes	Segmental	93.425
*SlIDD11*/*SlIDD4*	0.325	2.007	0.162	Yes	Segmental	66.906
*SlIDD11/SlIDD8*	0.201	1.233	0.163	Yes	Segmental	41.090
*SlIDD11*/*SlIDD10*	0.373	1.691	0.220	Yes	Segmental	56.361
*SlIDD4*/*SlIDD10*	0.229	0.924	0.248	Yes	Segmental	30.816
*SlIDD4*/*SlIDD8*	0.311	1.675	0.186	Yes	Segmental	55.819
*SlIDD17*/*SlIDD16*	0.162	0.777	0.208	Yes	Segmental	25.896
*SlIDD16*/*SlIDD6*	0.361	1.917	0.188	Yes	Segmental	63.900

## Data Availability

The data that support the findings of this study are available in the Appendix A of this article.

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
