# Peer review of "Genome-Wide Characterization of the INDETERMINATE DOMAIN (IDD) Zinc Finger Gene Family in Solanum lycopersicum and the Functional Analysis of SlIDD15 in Shoot Gravitropism"

_ijms, 2024, doi:10.3390/ijms251910422_

Round 1

Reviewer 1 Report

Comments and Suggestions for Authors

In this study, Wu et al. isolated 17 IDD genes from tomato and performed a functional nanlysis of IDD15. I think this work is interesting for providing the information of IDD gene family in tomato. I just have some comments as below:

1.Line 45, 'The first gene identified in the IDD family of transcription factors was ZmID1' should be 'The first member identified in the IDD transcription factor family was ZmID1'.

2.Line 52, 53, genes Hd3a, RFT1 shoud be pointed out the full name. Please check if all gene names are given out the full names.

3.L115.  why you only use ZmID1 to confirm the order of SlIDDs.

4. For example, gene names in Table 1 shoud be Italic. Thus, check this problem in all figures and Tables.

5.What's meaning of the scale bar in Fig. 4a?

6.Please change the background color of Fig. 5, I can not see the numbers of each elements.

7.In Fig. 6, MG means mature green? If so, please change MG, B, B10 into their names as the some as figure legend.

8. In Fig. 7, please do differential analysis of qRT-PCR results. The results showed the 'relative expression of SlIDD genes', so the expression level in which tissue was set as 1?

9. Line 359 and Fig. 9b, how many bp of deletion in M2? 5 bp or 7 bp?

10. I don't know if the negative results of IDD15 could be accepted by IJMS or not. In my opinon, it is meaningless result to tell readers that IDD15 has no relationship with 'Shoot Gravitropism'. I recommend you to do more observations of other indexes such as plant height, flowering time, yield, and so on.

Author Response

Dear Reviewer:

Thank you very much for reviewing our manuscript, and we really appreciate these helpful points raised by the reviewers and editorial board.

According to these suggestions from reviewers and editorial board, we have modified the manuscript point by point in the revised manuscript. In summary, the writing of the manuscript has been polished; and these figures and legends have been modified. The method has been described more adequately; and the results have been presented clearer. For clarity, all comments are given with black letters and responses are given with the blue letters below. All comments and suggestions have been replied one by one and addressed in the revision.

Sincerely,

Jianghua Chen (Corresponding author)

Quanzi Bai (Corresponding author)

Reviewer 2 Report

Comments and Suggestions for Authors

Major revision

Author Response

(The authors gave the same response as above.)

Round 2

Reviewer 1 Report

Comments and Suggestions for Authors

I have no more suggestions for this verison of MS.